# Research on Urban Fire Station Layout Planning Based on a Combined Model Method

**Zhijin Yu, Lan Xu \*, Shuangshuang Chen and Ce Jin**

College of Safety Science and Engineering, Yanta Campus, Xi'an University of Science and Technology, Xi'an 710054, China
* Correspondence: 20220226108@stu.xust.edu.cn

**Abstract:** With the rapid development of urbanization, fire risk factors have increased greatly, indicating a higher requirement for urban firefighting security. Fire rescue capabilities can be effectively improved by the scientific layout of fire stations, and therefore, the optimal spatial arrangement of fire stations has practical implications for urban safety. In this paper, a method for planning the locations of urban fire stations is presented, taking into account the fire risk points of interest (POIs) data, road networks and fire station planning principles. The combined model method is validated against the nearest facility point model, and the service area model is proposed for the coverage of POIs and regional areas of planned new sites. The efficacy of the model is demonstrated through an improvement in the coverage of crosspoints of the regional area and points of interest (POIs), with increases of 10.20% and 12.43%, respectively. We applied the combined model method to Fengdong New Town, Shaanxi Province, China. A total of 11 new potential sites were proposed to improve the efficiency of spatial coverage, and subsequently, the coverage rate of the POIs and regional area reached 97.66% and 84.80%, respectively. This study provides application guidelines for the decision-making of fire services and the allocation of firefighting resources.

**Keywords:** urban firefighting security; fire rescue capability; combined model method; fire station planning; location–allocation model; maximal covering location mode

## 1. Introduction

In recent years, potential fire risks have been ever-increasing due to the deepening conflict between urban population densities and inadequate firefighting resource allocation. Frequent fires pose direct or indirect threats to human life, and seriously affect economic development. From 2013 to 2016, China experienced record-high fire rates, with over 300,000 incidents reported annually. Urban fires were responsible for nearly 60% of the total incidents [1]. Therefore, enhancing urban fire rescue forces is of great significance in reducing economic losses and casualties.

Fire stations and firefighters are the backbone of rescue capabilities in urban communities. The key strategy to improve emergency response capabilities is to enhance the spatial accessibility to fire scenes. In most Chinese cities, fire stations are located using the "circle-drawing method" (drawing a circle with an area of 7 km$^2$ or 15 km$^2$ centered on each fire station as the service area). This method usually leads to excessive overlap of jurisdiction areas, and this kind of arrangement mainly considers the linear distance traveled by the fire vehicle. However, other factors, such as population density, road traffic, building density, geographic position, etc. are easily overlooked, leading to an increase in the possibility of not responding to a call for service in a timely manner. There is a certain gap between the urban firefighting capacity and the actual needs of urban development. Based on this gap, the aim of this study was to consider the factors of fire station layout comprehensively and to enhance the efficiency of urban fire response capabilities and the coverage of high-risk places. The response time from the fire station to the fire scene

mainly depends on the number and locations of fire stations, i.e., whether the layout of the fire stations is reasonable [2–4]. Research on facility location problems appears to have begun in 1909 when Weber [5] discussed the siting of facilities in industrial zones, and then in 1963 when Cooler extended the industrial zone theory and proposed the location–allocation (LA) model. At the end of the 20th century, Badenso and Owen proposed the P-center model and the location set coverage problem (LSCP) model [6,7]. In recent years, the research on fire station siting has gradually deepened. Scholars have proposed the maximal covering location problem (MCLP) for the locational set coverage model (LSCP) and the maximal covering location model [8–11]. Such studies have often started from the coverage area of fire stations and maximized the coverage of fire stations through various mathematical models. However, this method leads to averaging the firefighting demand between high-risk and low-risk areas, increasing the randomness of fire station service area divisions and the decentralized distribution of fire stations. Wang and Sirbiladze et al. developed and generalized a multi-objective fuzzy model to solve the problem of fire station siting in a fuzzy environment [12–14]. Murray performed discrete optimization of the model [15] from an economic perspective, and Ming et al. proposed a distributed robust model considering the uncertainty of the rescue time [16]. The aforementioned studies have considered various fire-initiating risk factors for the planning of various fire stations, but have not fully incorporated spatial variability into the changing demand for fire services. Therefore, it is important to develop viable methods to reveal the relationships among the number of facilities, the terrain, and fire rescue demand.

With the continuous development and innovation of geographic technology, new data sources and tool geospatial technologies support a more accurate and quantitative perception of fire safety posture in cities. In urban fire planning, the layout of fire stations is often determined by the fire risk distribution in urban areas and the service areas of fire stations [17–19]. The distribution of fire risk areas can be quantitatively identified by the POI location data, which is helpful for solving the problem of insufficient targeting of fire station siting. Currently, POI data are mainly used in urban planning, design [20–22], etc., and is now gradually being applied to urban security. Wang et al. used POIs and multi-temporal traffic condition (MTS) data to study the actual coverage rate of fire stations under different traffic conditions in central Beijing [23]. Wang et al. determined the distribution of fire risk by performing kernel density analysis (KDA) on all categories of POI using ArcGIS software [24]. These explorations have provided an effective tool for comprehensively evaluating the location of fire stations based on the geographical features of the urban environment on a citywide scale. Emergency response time is also a key factor that determines the effectiveness of fire rescue. Considering that the efficiency of firefighting response time for accident rescue operations is crucial for their impact on accident losses [25], Rodriguez et al. proposed a technical model for facility siting and equipment seating with required coverage [26,27]. Han et al. combined the spatial distribution of urban fire risk with ArcGIS spatial analysis techniques to reduce the time for firefighting forces to reach high-risk sites [28]. Yu et al. introduced the variable of fire truck travel time in the fire station siting process, and Boschetti et al. proposed a metaheuristic algorithm based on the ensemble coverage problem [29,30]. The actual response time of a fire station is closely related to traffic conditions and the distance traveled by fire trucks in a rescue [31]. Meanwhile, mixed traffic conditions on urban roads often lead to widespread traffic congestion [32,33]. To address the issue of traffic obstruction encountered by fire trucks during travel, Liu focused on a joint location and assignment problem, while Zhang introduced the barrier-free road model, which has been shown to improve travel efficiency in studies [34,35].

As previously mentioned, scholars have made considerable progress in determining the locations of urban fire stations through the development of various models. However, with the growing significance of urban fire incidents, there are important considerations to be made regarding these models. Firstly, the most current fire station location models only take into account factors such as emergency vehicle traffic flow, population density,

and fire occurrence history, and do not adequately consider the coverage of high-risk fire areas. These areas, if not adequately covered, can result in substantial property damage and casualties. To ensure prompt emergency response in urban areas, it is critical to optimize the fire station layout for optimal coverage. Secondly, the construction of urban fire stations is a long-term process, and the model must also consider future planning based on the existing fire stations in the city. Disregarding this consideration may result in decreased coverage efficiency and the incurrence of unnecessary socio-economic costs. Therefore, it is imperative to plan the fire station layout and improve coverage in high-risk fire areas, taking into account existing urban construction. Accurate geographic data on fire stations and fire risk areas are fundamental for establishing a solid foundation for the model and selecting suitable fire station sites.

In this study, a fire station layout method based on the location–allocation model is proposed. The aim of this study was to determine the optimal fire station locations, taking into account the existing fire stations to solve the issue of strategically siting future fire stations in a city. Although the locations of fire stations have been well studied, to the best of our knowledge, there have been few studies focusing on the impact of fire stations that have been constructed on the planned layout of fire stations that will be constructed in the future. Among all improvement opportunities, no scientific investigation has been undertaken regarding the sitings of Fengdong New Town fire brigades; this study explores this gap by using GIS-based fire station location optimization models. The implications of this research are significant for both China and other countries around the world. In China, where rapid urbanization and population growth are putting increasing pressure on fire stations to respond quickly and efficiently to emergencies, our findings could inform the strategic sitings of new fire stations to improve public safety. It is important to note that the location–allocation model used in this study was developed by the authors specifically for this research project and was adapted to the setting of Fengdong New Town.

## 2. Methodology

### 2.1. Study Area

The study area was Fengdong New Town in Xi'an City, China, which is an important part of the South Bank of Weihe River in Xixian New Area. It is the region with the highest GDP and the largest population in Xixian New District, located at $108°47'15''$ E and $34°15'8''$ N. The area covers almost 159.3 km$^2$ and comprises 35 communities and 107 administrative villages. From 2016 to 2019, there were 437 fires in Fengdong New City, resulting in 2 casualties and direct property losses of about CNY 4,900,000. Electrical fires were the most prevalent incidents, accounting for 33.2% of the total number of fires, followed by those caused by smoking and careless use of fire, which accounted for 13.6% and 6%, respectively. In recent years, Fengdong New City has undergone rapid development, leading to an increase in the number and types of medium-high-risk fire units and resulting in frequent occurrences of "small-scale fire fatality" incidents.

By 2021, 4 fire stations were built (including 2 special fire stations, 1 general class I fire station, and 2 general class II fire stations). However, the number of existing fire stations falls short of the actual demand. Figure 1 shows the spatial and locational information of Fengdong New Town.

### 2.2. Data Source

The geospatial data used in this study were obtained through data collection and field research. Specifically, we mainly sourced the following four types of data from publicly available online repositories or websites.

(1)  Point data processing

The point data required for this study included building points, existing fire station points, and the geospatial locations of medium-high-risk quantifiable indicators in the study area.

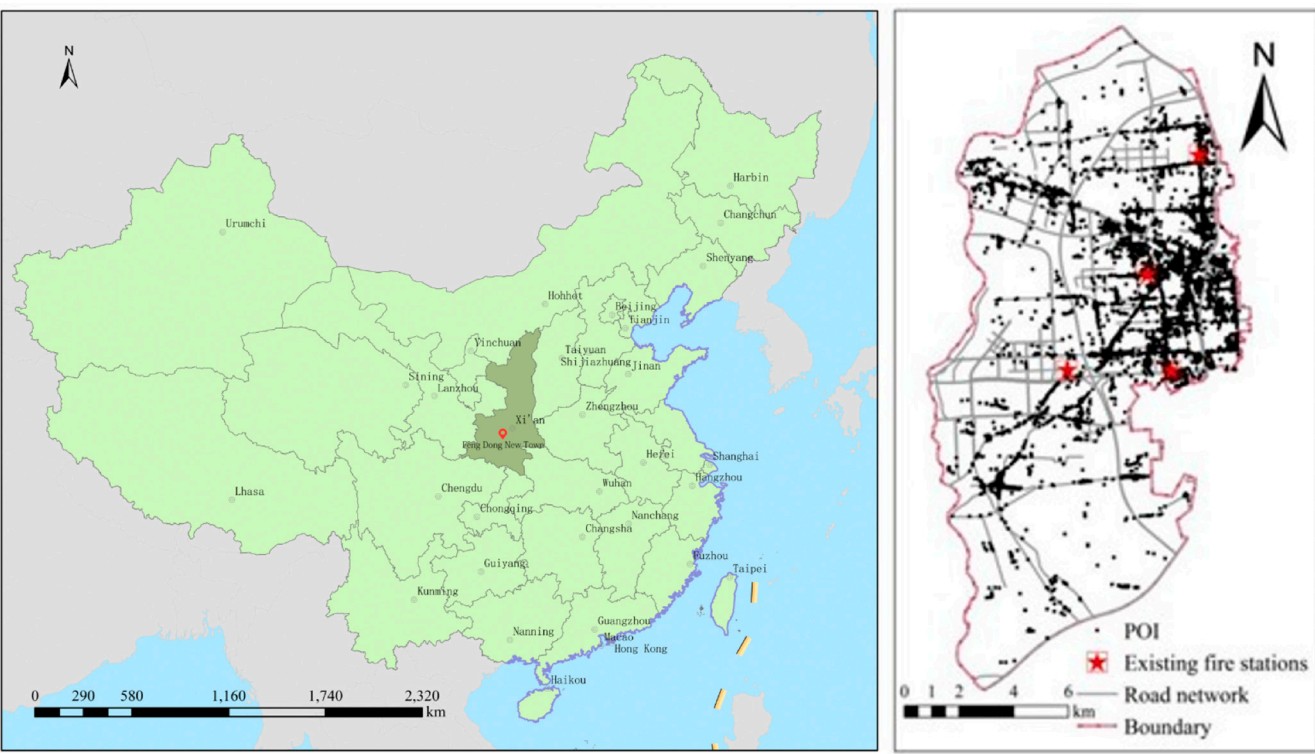

**Figure 1.** The geographical location of Fengdong New Town and the locations and spatial distribution of fire stations in the region.

① POI: Points of interest (POIs) are an important part of geographic information, and mainly contain 3 attributes: name, category, and spatial location. These represent real restaurants, hospitals, shopping places, schools, tourist attractions and other buildings or regional places in the form of geospatial points. The POIs in this study were all from the online Baidu Open Map (https://lbsyun.baidu.com/ (accessed on 20 November 2020)). A total of 13,073 valid data were included after coordinate conversion, screening, and sorting, with an average distribution density of 64 data/km². The classification and quantity of POIs are shown in Table 1.

② Existing fire station locations: In order to ensure the position data, in this study, the existing fire station spatial locations of Fengdong New Town were extracted from the Gaode Map (https://www.amap.com/ (accessed on 20 November 2020)) and the Baidu Map (https://map.baid.com/ (accessed on 20 November 2020)), and after data de-duplication, merging, and elimination, the fire station spatial locations were obtained, as shown in Figure 1.

③ Spatial location of quantifiable indicators: The spatial locations of medium-high-risk quantifiable indicators were extracted based on the POIs.

**Table 1.** Sorted POIs of Fengdong New Town.

| Type | Quantity ($p$) | Type | Quantity ($p$) | Type | Quantity ($p$) |
|---|---|---|---|---|---|
| Catering industry | 2854 | Traffic facility | 192 | Life service | 2380 |
| Scenic spots | 55 | Education | 549 | Gymnastics | 282 |
| Hospitality | 538 | Finance | 98 | Medical treatment | 444 |
| Shopping | 4946 | Commercial housing | 362 | Government office | 373 |

(2) Line data processing

Line data are the data of "connection" in accessibility analyses, and the main line data required in this study were the road network data of the study area, obtained from

OpenStreetMap ((https://www.openstreetmap.org/ (accessed on 20 November 2020)). Based on the actual road distribution in Fengdong New Town, the road network data were obtained by error checking and editing and modifying the road network. The spatial locations are shown in Figure 1.

(3)   Surface data processing

The fire station layout optimization process involves areas occupied by some buildings that are larger than the buffer areas of the POIs. To ensure the accuracy of the study results, the actual footprints of these parts of the buildings, i.e., the shapes of the building spaces, were extracted. In addition, the study area in the research process was essential.

① Study area: As displayed in Figure 1, the study area was extracted from OpenStreetMap (https://www.openstreetmap.org/ (accessed on 20 November 2020)), an open road map database.

② The occupied area of buildings: Actual measurements of Fengdong New Town were used to determine the approximate spatial locations and area size of buildings. Finally, we used the open road map database OpenStreetMap (https://www.openstreetmap.org/ (accessed on 20 November 2020)) to extract the actual footprints and spatial locations of these buildings.

(4)   Three-dimensional data processing

After downloading the digital elevation model (DEM) data of Fengdong New Town from the geospatial data cloud, we transformed these data into a three-dimensional TIN using ArcGIS. Areas with an elevation of less than 60 m and slope not greater than 30 degrees were taken as the buildable areas for fire stations based on slope direction analysis.

*2.3. Model Support*

There is a large amount of geographic data available for cities, consisting of numerous randomly distributed data points. These data points must be incorporated into a network model, which is then transformed into a location set cover model subject to the restrictions imposed by land allocation. This allows for the resolution of the location selection problem. Given these characteristics, several fire station location models have been proposed. Of these models, the following four are commonly used.

(1)   Maximal Covering Location Model

The objective of this model is to maximize the total demand covered by the sited facility points. All demand points are assigned to the located facility points as much as possible under the specified constraint conditions.

The specific objective function and constraint conditions of this model are shown in Equation (2).

Objective function:

$$Max[\sum_{i \in I}^{m} Z_i] \tag{1}$$

Constraint conditions:

$$Z_i - \sum_{j=N_i} X_j \leq 0 \ \forall i \in I, \ \forall j \in J$$

$$\sum_{j \in J}^{m} X_j = P \ \forall j \in J$$

$$X_j \in \{0,1\} \ \forall j \in J$$

$$Z_i \in \{0,1\} \ \forall i \in I$$

$I$—Demand points, $I = 1, 2, 3, \ldots \ldots , n$;
$J$—Facility points, $J = 1, 2, 3, \ldots \ldots , m$;
$Z_i$—The $i$-th demand point covered;

$X_j$—Location supply point for facility point *j*;
$N_i$—A collection of fire station alternatives that can serve a point of need;
*P*—Number of supply points.

(2)    Minimum Facility Point Model

The objective of this model is to minimize the number of facility points needed within the specified impedance while satisfying as many demand points as possible.

The specific objective function and constraint conditions of this model are shown in Equation (2).

Objective function:

$$Min[\sum_{j=1}^{n} X_j] \tag{2}$$

Constraint conditions:

$$\sum_{j=N_i}^{m} X_j \geq 1 \ \forall i \in I, \ \forall j \in J$$

$$X_j \in \{0,1\} \forall j \in J$$

*I*—Demand points, *I* = 1, 2, 3, . . . . . . , *n*;
*J*—Facility points, *J* = 1, 2, 3, . . . . . . , *m*;
$X_j$—Location supply point for facility point *j*;
$N_i$—A collection of fire station alternatives that can serve a point of need.

(3)    Nearest Facility Model

The nearest facility point model is also referred to as the P-center problem. After determining the number of service facility points, the model measures the cost of the facility point to reach the event point within a certain time or distance impedance. This is applied to determine the shortest trip from the facility point to the demand point, minimizing the maximum distance between all emergency points and their service facilities. The coverage of the facility point to the demand point can be also determined by this model.

(4)    Service Area Model

A service area is the amount of land, people, or other demand points in the area. This area covers existing streets within the specified constraint conditions and is used to determine the facility site. In addition, the service area is often used to assess the accessibility of a facility point. As shown in Figure 1, since the final simulation results of the model show the coverage of the area by the facility point within a certain impedance, the service area model can be used to determine the coverage of an area by a facility point.

*2.4. Workflow of Research Methodology*

In this study, an approach was developed based on the integration of the maximal covering location model and the minimum facility point model. The optimal spatial locations and configurations of fire stations were obtained from this combined model method. First, the land resources, points of interest (POIs), slope, and other relevant factors were quantified. Then, the buffer analysis and overlay analysis functions in ArcGIS were utilized to identify suitable areas for constructing fire stations. Secondly, a buffer analysis was conducted on the existing fire stations in the study area to determine the first group of candidate fire station points. Subsequently, the time impedance and distance impedance were established using ArcGIS and the maximized coverage model and minimized facility point model, and the crosspoints of the two models were taken as the required points for the first group of fire stations. The coverage efficiency of the first group of candidate points and the existing fire stations in the city were then evaluated using the service area model and nearest facility point model. If the coverage ratio of the POIs to the area did not meet the desired criteria, the buffer zone analysis was repeated on the first group of fire station candidates and existing fire stations. This was carried out to further identify

the fire station candidates. The maximization coverage model and minimization facility point model were re-imported, and the final fire station points were established once the resulting fire stations met the urban fire prevention and control requirements. A flowchart of the proposed method is presented in Figure 2.

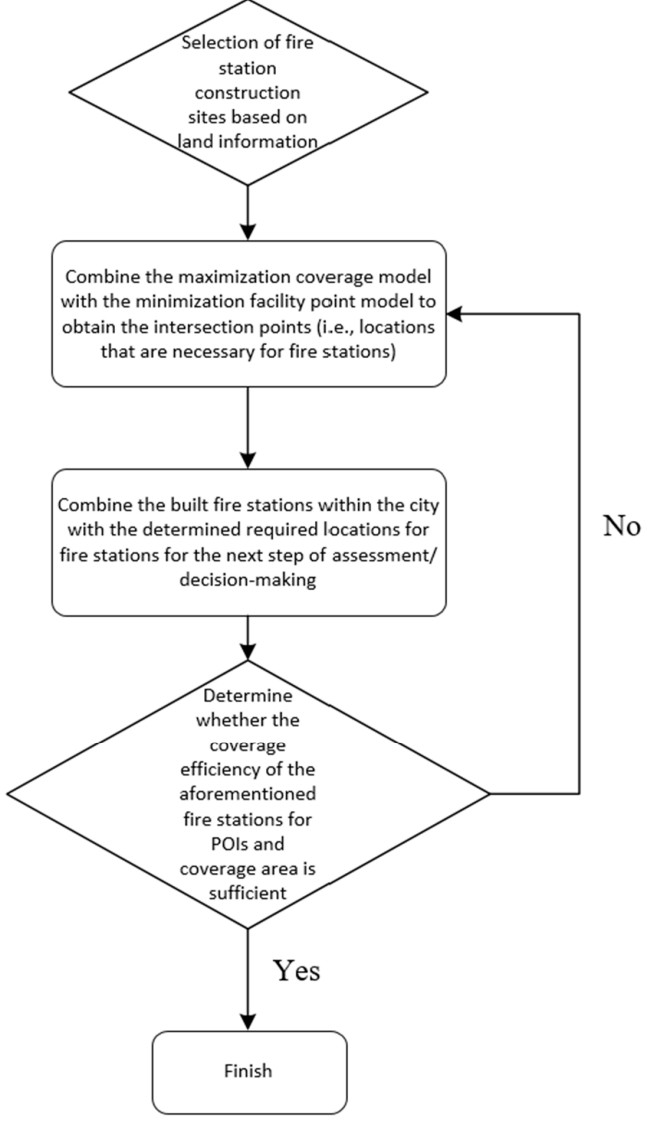

**Figure 2.** Flowchart of the proposed method.

## 3. Result

### 3.1. Method Test

#### 3.1.1. Layout Principles [36]

(1) Fire station precinct area: According to China's national standard code for urban fire control planning, the protection area of fire station precincts varies by level: for class I fire stations, it is no more than 7 km$^2$; for class II fire stations, it is no more than 4 km$^2$. The precinct areas of special branches of fire stations, which also have firefighting and rescue tasks within the precinct, are similar to that of primary stations. Therefore, based on this regulation, we set the buffer values for city fire stations to 4 km$^2$ and 7 km$^2$.

(2) Fire station construction land area: The index of fire station construction land area is a key indicator for fire station planning and construction. As stipulated in China's national standard code for urban fire control planning, the basic construction land

area of primary fire stations is 3900–5600 m$^2$. According to this construction standard, it is possible to screen the buildable area of fire stations in the study area.

(3)  Time principle: The layout of the fire stations should follow a general principle. The fire brigade can reach the edge of the precinct within 5 min after receiving the dispatch command. This 5 min period includes 1 min of preparation time and 4 min of travel time to the scene. This part specifies a vehicle travel time of 4 min for the fire station, so we next set the time impedance to 4 min.

### 3.1.2. Feasibility Analysis

In order to verify the validity of optimizing the fire station layout using the combined model method, the Chinese town of Fengdong was used as a case study. The proposed station locations were determined by the combined model method, and the sites close to the existing fire stations were selected. Then, the simulation-derived stations were comprehensively compared with the coverage of POIs and area within the existing fire station sites using the service area model and the nearest facility point model.

Considering the principle of fire station layout, the buffer value of POI was set to 60 m, the buffer value of partial buildings was set to 50 m, and the buffer value of Fengdong New Town Road was set to 15 m. Considering the distance principle, the side length of fire station construction land area was set as (75 + 50) m, and a fishing net surface with 125 m × 125 m side lengths of the buildable area was established. Finally, 9646 fishing net surfaces and 7442 fire station candidate points (fishing net points) were obtained. The candidate sites of Fengdong New City fire stations are displayed in Figure 3.

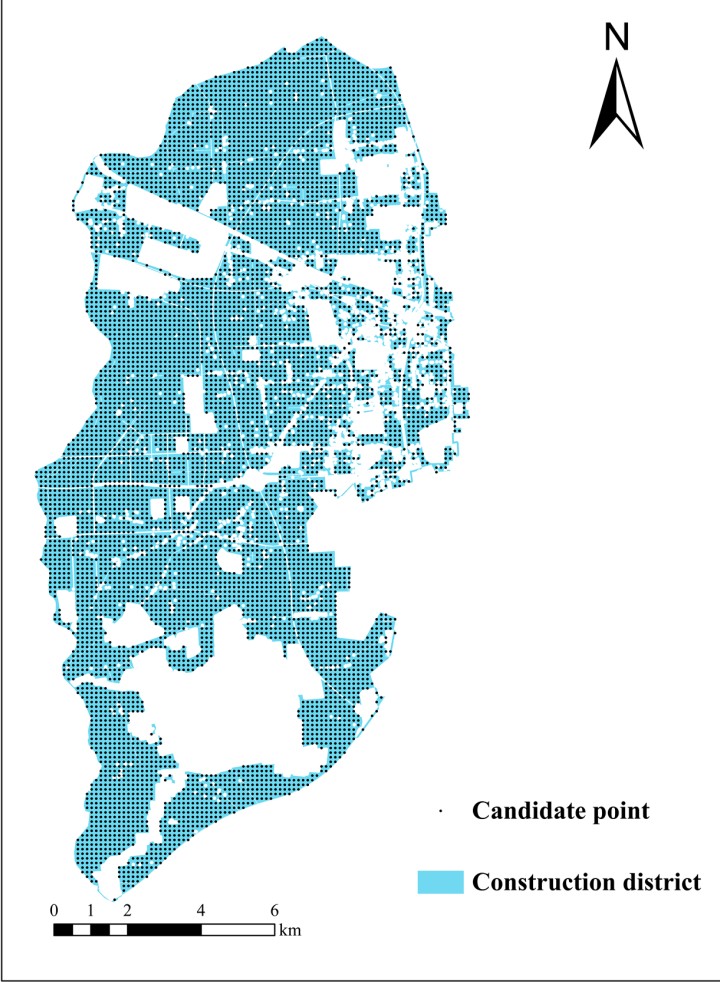

**Figure 3.** Spatial distribution of fire station candidate points.

Based on the buildable area for fire stations and the candidate fire station locations, a time impedance of 4 min was established, and the initial number of facility points was set to 23. The results obtained when the spatial location of the maximum coverage points was solved using the maximum coverage model are displayed in Figure 4a. With a maximum coverage point of 23, the coverage rate of POIs in Fengdong New Town reached 100%. The results obtained when the minimum facility point model was used to solve for the minimum number and spatial locations of facility points are shown in Figure 4b. With a minimum facility point of 17, the coverage rate of POIs in Fengdong New Town also reached 100%. However, it was observed that some of the resulting spatial points were not efficient. In order to guarantee the efficacy of the crosspoints, those with a POI coverage of less than 1% were omitted from the analysis. After screening, 12 maximum coverage points and 7 minimum facility points were obtained, resulting in POI coverage rates of 96.79% and 97.72% in Fengdong New Town, respectively, effectively achieving full coverage of the POIs in the Fengdong area.

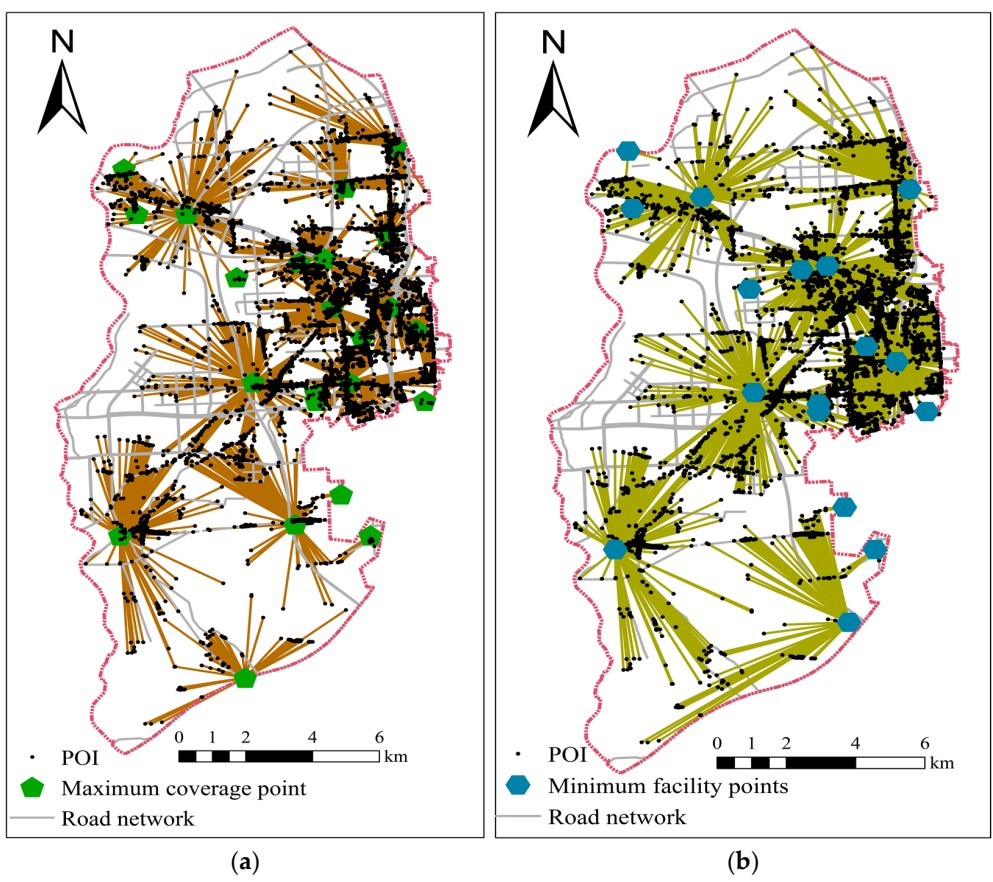

**Figure 4.** The spatial distribution results obtained from the simulation, (**a**) maximum coverage points; (**b**) minimum facility points.

It was found that the first result comprised three points. As depicted in Figure 5, the buffer zone was established using the area of the fire station's protection jurisdiction. Two of the crosspoints are located in proximity to two existing fire stations, named Nearest Point 1 and Nearest Point 2. The direct distance between Fire Station Point 1 and Nearest Point 2 is 1.08 km, while the direct distance between Fire Station Point 3 and Nearest Point 2 is 1.34 km. Subsequently, the nearest facility point model and service area model were employed to analyze the coverage of POIs and area, with a time impedance of 4 min set. The coverage of the existing fire stations and their adjacent points to the POIs and area of Fengdong New Town were then simulated in both single station and combined station scenarios. The comparative analysis results are presented in Figures 6 and 7.

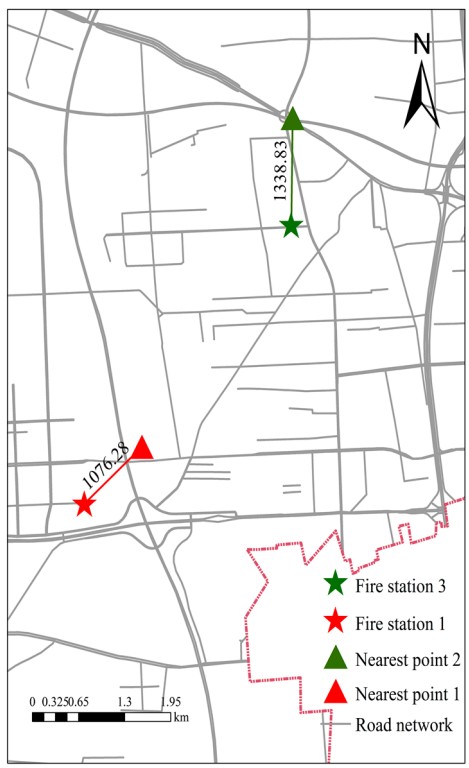

**Figure 5.** Distribution of fire stations and nearest points.

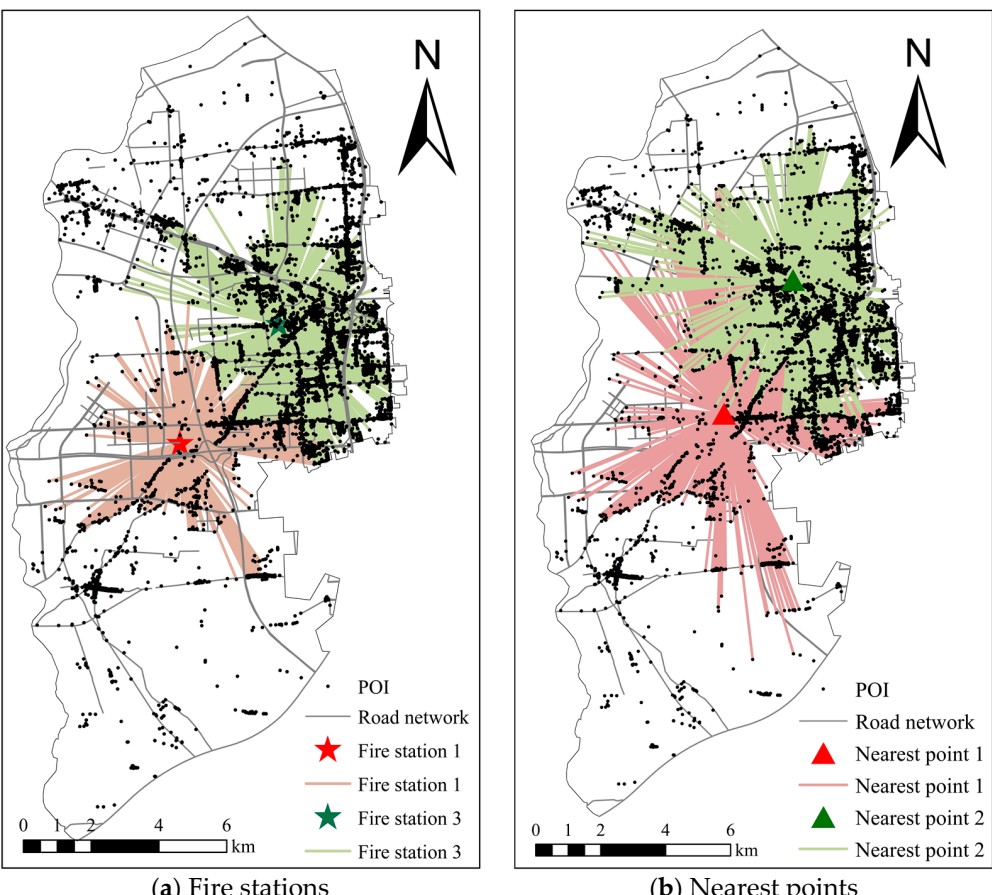

**Figure 6.** Coverage of POIs.

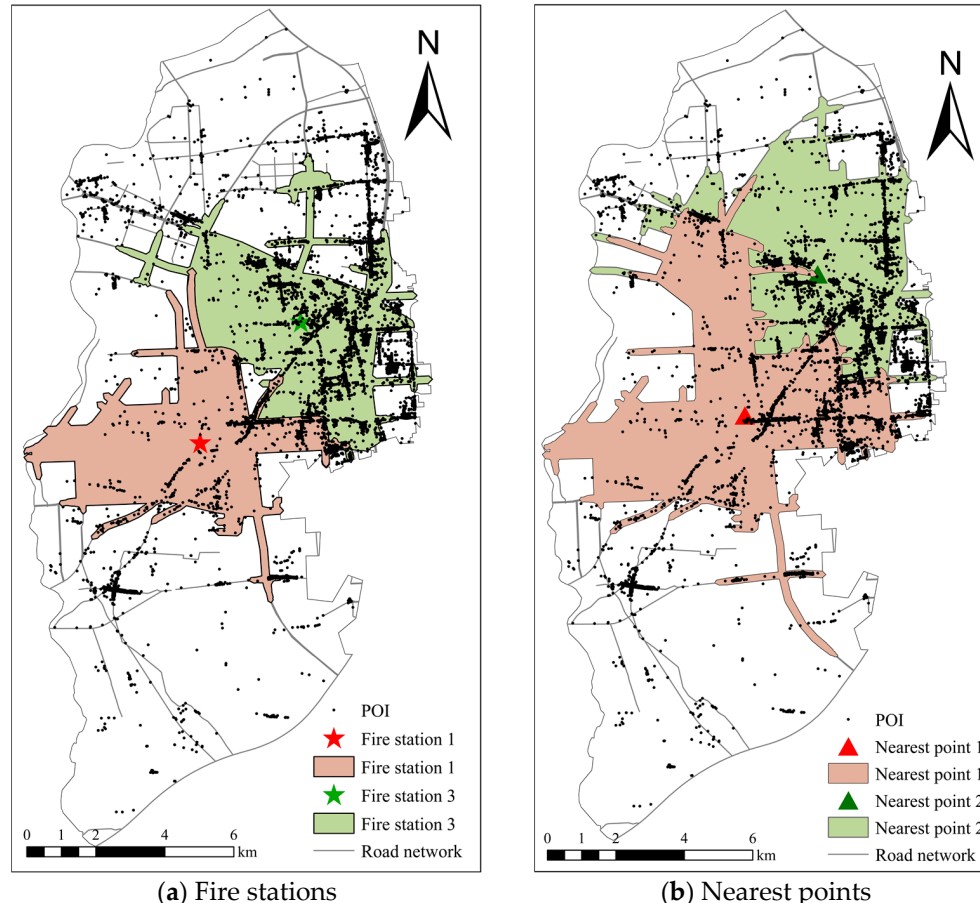

(**a**) Fire stations           (**b**) Nearest points

**Figure 7.** Coverage of area.

Table 2 shows that compared to the existing fire stations, the coverage rates of POIs and regional areas of Fengdong New Town increased by 17.90% and 11.34%, respectively, when using Nearest Point 1. While the coverage rate of POIs by Nearest Point 2 remained unchanged, the coverage rate of the regional area increased by 9.11%. After the merger of the stations, the coverage rate of the nearest points to the POIs and the regional area increased by 10.20% and 12.43%, respectively, compared to the existing fire station points. These findings suggest that the combined model method can not only optimize the location of existing fire station points but can also effectively plan the spatial layout of fire stations.

**Table 2.** Coverage of POIs and area by fire stations and nearest points.

| Site Type | Site Name | Linear Distance | POIs | | Regional Area | |
|---|---|---|---|---|---|---|
| | | | Coverage Amount (*p*) | Coverage Rate | Coverage Area (km²) | Coverage Rate |
| Single-node analysis | Fire Station 1 | 1076 m | 1599 | 15.65% | 25.97 | 16.30% |
| | Nearest Point 1 | | 3428 | 33.55% | 44.03 | 27.64% |
| | Fire Station 3 | 1338 m | 6765 | 66.20% | 29.45 | 18.49% |
| | Nearest Point 2 | | 6841 | 66.94% | 43.97 | 27.60% |
| Combined analysis | Fire Station | — | 7256 | 71.00% | 53.48 | 33.57% |
| | Nearest Point | — | 8298 | 81.20% | 73.28 | 46.00% |

*3.2. Layout Optimization of Fire Stations Based on Combined Model Method*

In the preceding discussion, the combined model method was analyzed. In this section, several methods are used to optimize the layout of urban fire stations in Fengdong New Town. To ensure minimal overlap in the precinct protection area between the planned and

existing fire stations and to promote cost-effectiveness, the linear distance from the primary fire station to the farthest points of the precinct was calculated using the formula $A = 2P^2$, where P represents the protection radius of the fire station, which was determined to be 1870.8 m. Next, a buffer analysis was performed on the precinct protection area of fire stations using P as the buffer value. The maximum allowable road speed in China, which is 60 km/h, was set as the fire truck travel speed. Based on this data, the buildable area of fire stations in Fengdong New Town and the distribution of candidate points for fire stations were determined.

The first application of the combined model method yielded six crosspoints, which are referred to as Crosspoint 1 in Figure 8. At this point, the coverage rate of points of interest (POIs) reached 96.56%, while the coverage rate of the regional area was only 66.92%. It was observed that some POIs and regions, particularly in the southwest (shown in white in the figure), were not covered. Further details are presented in Figure 8 and Table 3, which indicate that certain POIs and regions remained uncovered. To ensure that the POIs and regions of Fengdong New Town were covered as much as possible, Crosspoint 1 was selected as the demand point for the fire station. Using the combined model method, the remaining areas were further studied after regional screening of the needed and existing fire station points.

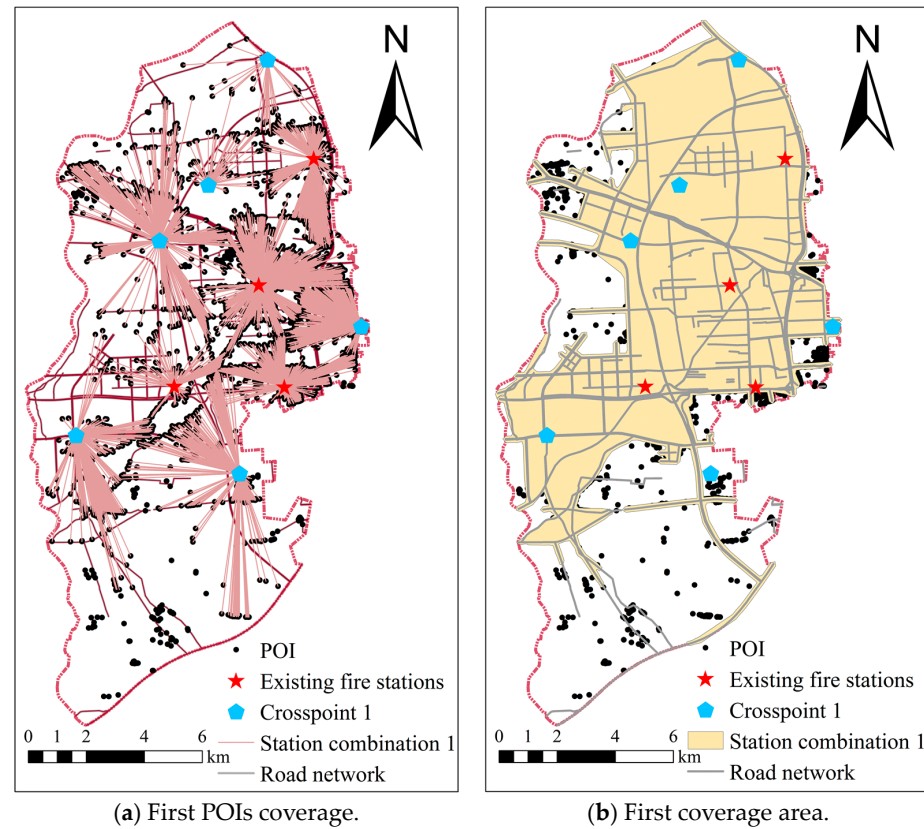

(**a**) First POIs coverage.    (**b**) First coverage area.

**Figure 8.** Results of the first fire station location simulation.

**Table 3.** Coverage of POI and area by Crosspoint 1.

| Site Type | Number of Sites (*p*) | POIs | | Regional Area | |
|---|---|---|---|---|---|
| | | Coverage Amount (*p*) | Coverage Rate | Coverage Area (km²) | Coverage Rate |
| Fire Station | 4 | 8236 | 80.59% | 74.57 | 46.81% |
| Crosspoint 1 | 6 | 9112 | 89.17% | 101.94 | 63.99% |
| Station Combination 1 | 10 | 9869 | 96.56% | 106.60 | 66.92% |

The site buffer area was larger than the regional area of Fengdong New Town after 4 iterations, the number of crosspoints was 12, and the coverage rate of POIs and the regional area remained similar to that of the third iteration, as shown in Table 4. Compared to the second iteration, the POIs and the regional area obtained from the third iteration of the site layout increased by 0.06% and 0.9%, respectively, indicating a small upward trend. Considering the future firefighting development plan of the city, the optimal results for fire station site selection were achieved with plans for 11 new fire stations in Fengdong New City based on the calculation results of the first three iterations. At present, the coverage rates of the POIs and the study area reached 97.66% and 84.80%, respectively. The specific distribution of the planned stations is illustrated in Figure 9.

**Table 4.** Coverage of POIs and area by crosspoints.

| Site Type | Number of Sites ($p$) | POIs | | Regional Area | |
|---|---|---|---|---|---|
| | | Coverage ($p$) | Coverage Rate | Coverage Amount (km²) | Coverage Rate |
| Fire Station | 4 | 8236 | 80.59% | 74.57 | 46.81% |
| Crosspoint 1 | 6 | 9112 | 89.17% | 101.94 | 63.99% |
| Crosspoint 2 | 3 | 1843 | 18.04% | 54.45 | 34.18% |
| Crosspoint 3 | 2 | 1843 | 18.04% | 54.45 | 34.18% |
| Crosspoint 4 | 1 | 30 | 0.29% | 5.36 | 3.36% |
| Station Combination 1 | 10 | 9869 | 96.56% | 106.60 | 66.92% |
| Station Combination 2 | 13 | 9974 | 97.60% | 133.65 | 83.90% |
| Station Combination 3 | 15 | 9980 | 97.66% | 135.08 | 84.80% |
| Station Combination 4 | 16 | 9980 | 97.66% | 135.36 | 84.97% |

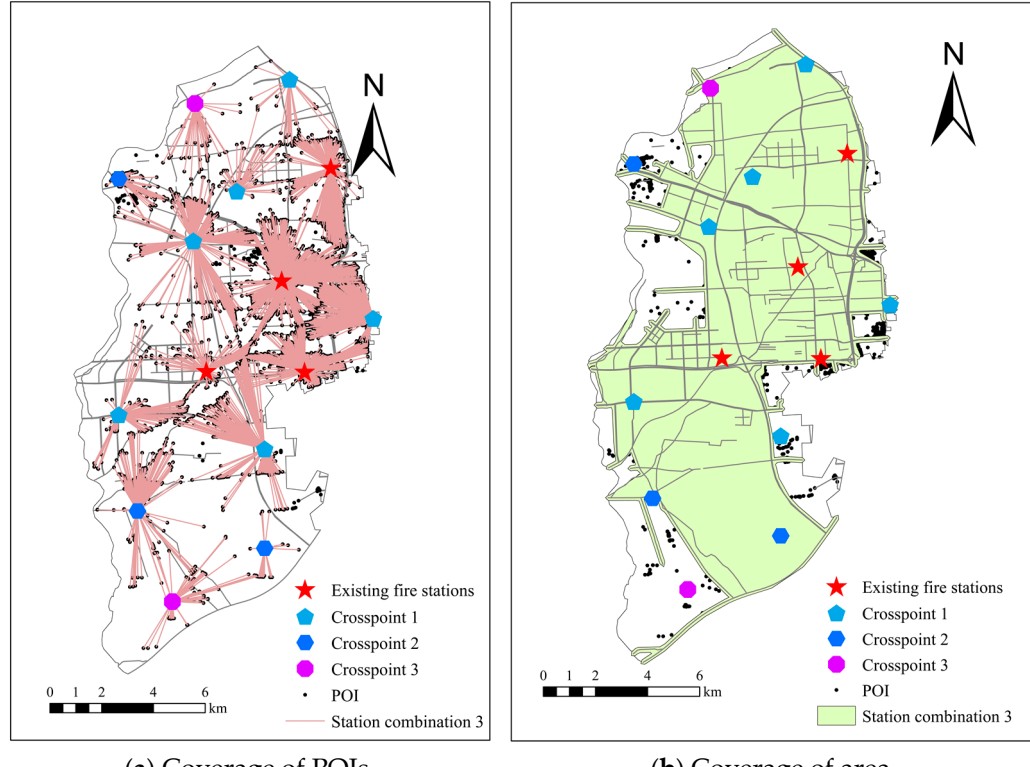

(**a**) Coverage of POIs.　　　　　(**b**) Coverage of area.

**Figure 9.** Final fire station location simulation results.

As presented in Figures 10 and 11, the results demonstrate the impact of adding fire station points on a 4 min firefighting travel time. The amount of risk indicator coverage was simulated at 1 min, 2 min, and 3 min intervals. As seen in Tables 5 and 6, the average

increase in coverage for POIs was 19.76%, and the average increase in area coverage was 35.78%. The maximum differences in coverage between POIs and areas were observed at 2 and 3 min, reaching 26.28% and 49.38%, respectively. The coverage rates of POIs and regional areas at 3 min for the additional stations were found to be virtually identical to the coverage rates at 4 min, with a difference of approximately 1%.

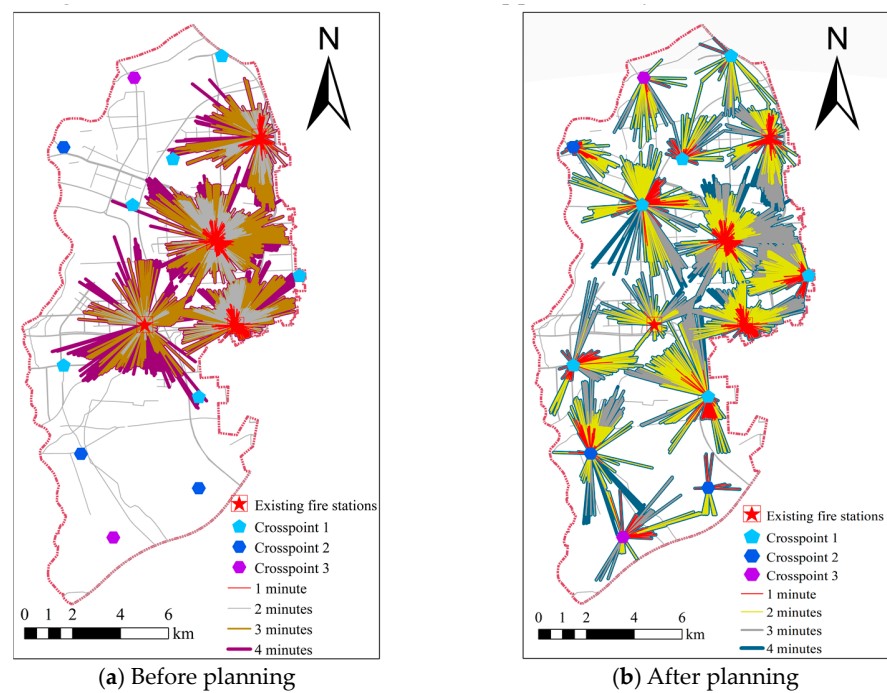

**Figure 10.** POI coverage in different time periods.

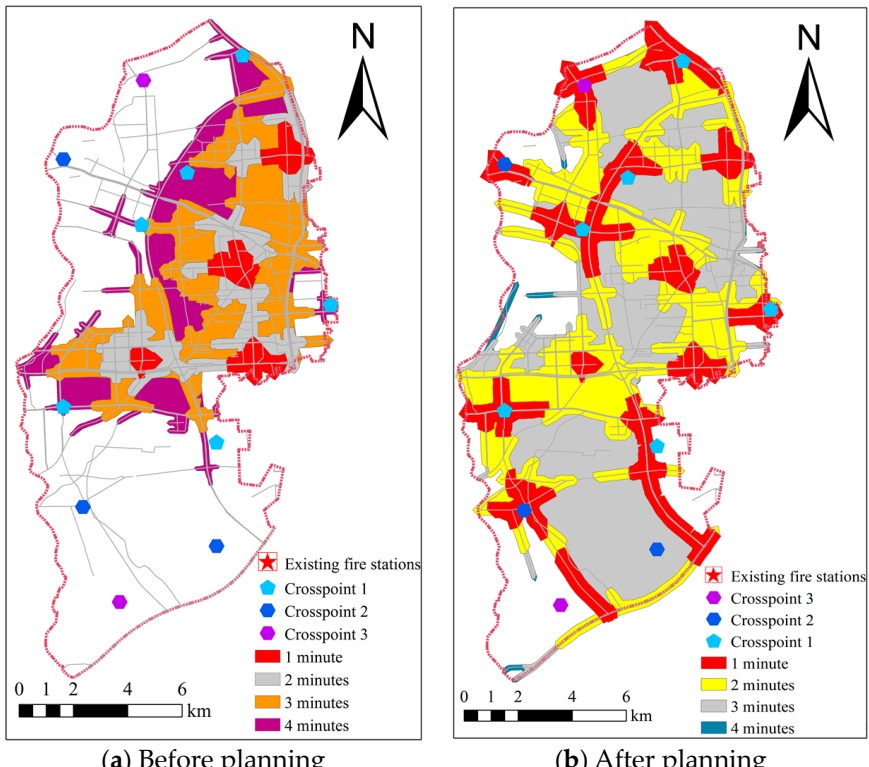

**Figure 11.** Area coverage in different time periods.

**Table 5.** Coverage of POIs in different time periods before and after planning.

| Site Type | 1 min | | 2 min | | 3 min | | 4 min | |
| --- | --- | --- | --- | --- | --- | --- | --- | --- |
| | Coverage Amount (*p*) | Coverage Rate | Coverage Amount (*p*) | Coverage Rate | Coverage Amount (*p*) | Coverage Rate | Coverage Amount (*p*) | Coverage Rate |
| Before planning | 1414 | 10.82% | 5238 | 40.07% | 9222 | 70.54% | 10,140 | 77.56% |
| After planning | 2325 | 17.78% | 8673 | 66.34% | 12596 | 96.35% | 12,755 | 97.57% |

**Table 6.** Coverage area in different time periods before and after planning.

| Site Type | 1 min | | 2 min | | 3 min | | 4 min | |
| --- | --- | --- | --- | --- | --- | --- | --- | --- |
| | Coverage Area ($km^2$) | Coverage Rate | Coverage Area ($km^2$) | Coverage Rate | Coverage Area ($km^2$) | Coverage Rate | Coverage Area ($km^2$) | Coverage Rate |
| Before planning | 7.14 | 4.48% | 22.79 | 14.31% | 55.73 | 34.98% | 74.57 | 46.81% |
| After planning | 36.50 | 22.91% | 82.24 | 51.63% | 134.39 | 84.36% | 135.08 | 84.80% |

The investigation of risk places in Fengdong New Town revealed 185 high-risk areas, including 50-year-old neighborhoods, 53 high-risk buildings, 45 urban villages, 25 flammable and explosive locations (refueling/gas/methanol stations), and 12 commercial complexes. After extracting the points of interest (POIs) for these areas, the coverage rates of the simulated fire station layout for high-risk areas were calculated for four time periods, and the results are shown in Figure 12. As indicated in Table 7, when only considering the high-risk places identified in the research, the coverage rate of high-risk places achieved after planning exceeded 90% when the driving time for firefighting vehicles was 3 min. The point coverage rate and surface coverage rate within a 4 min driving time reached 98.4% and 96.8%, respectively, essentially achieving full coverage of high-risk areas within the specified time.

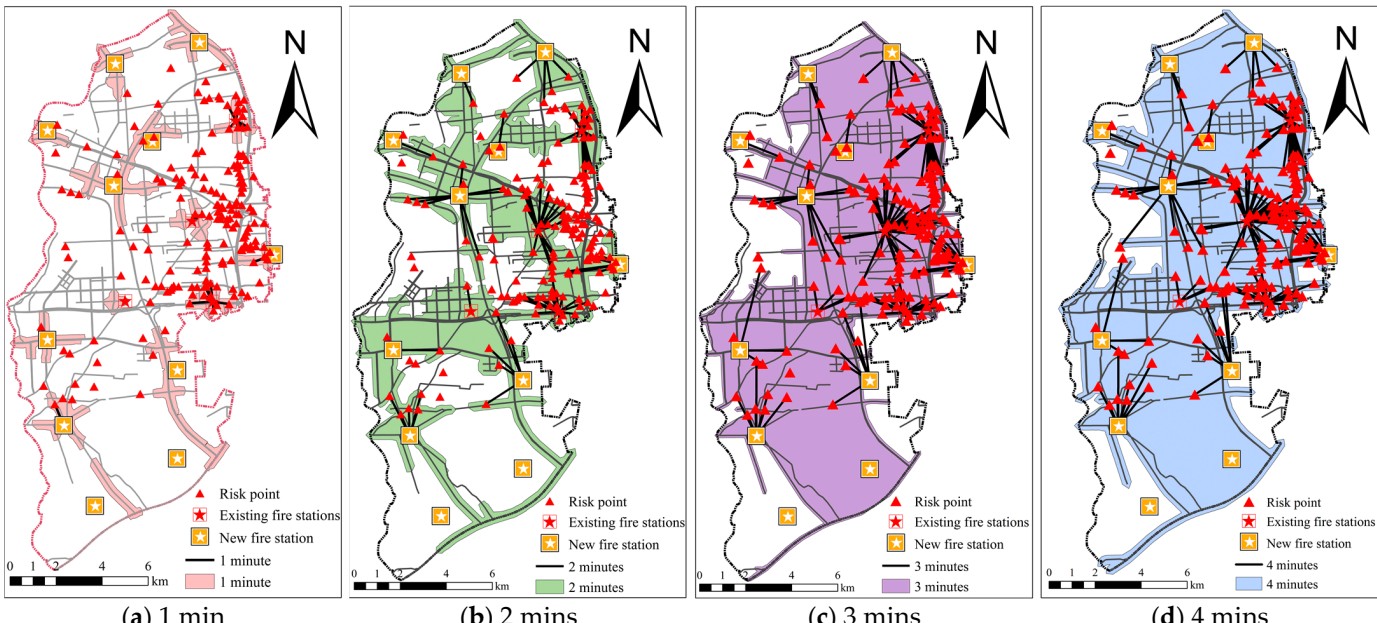

**Figure 12.** Coverage at each time.

**Table 7.** Coverage at each time.

| Risk Point | Risk Point | 1 min | | 2 min | | 3 min | | 4 min | |
|---|---|---|---|---|---|---|---|---|---|
| | | Amount (*p*) | Coverage Rate | Amount (*p*) | Coverage Rate | Amount (*p*) | Coverage Rate | Amount (*p*) | Coverage Rate |
| Point coverage | 185 | 30 | 16.2% | 108 | 58.4% | 175 | 94.6% | 182 | 98.4% |
| Surface coverage | | 27 | 14.6% | 112 | 60.5% | 173 | 93.5% | 179 | 96.8% |

## 4. Discussion

The above study provides a method for fire departments and decision-makers to choose a more appropriate pattern for the location of the new fire stations. As pointed out by Wang [3], the model with a single objective tends to neglect other requirements in fire emergencies, and the maximal covering location model usually leads to overlapping coverage. Moreover, in most of the earlier siting coverage models, when the emergency facility is within a preset value from the demand point, it is assumed that it can cover this demand point; otherwise, it cannot cover this demand point. However, although this assumption is theoretically feasible, it may result in an underestimation of the emergency system effectiveness in an actual emergency response. Therefore, in order to solve the problem of the inefficiency of the single coverage model, in the case study of Fengdong New Town, we combined the maximal covering location model with the minimum facility model. The combination of the two models maximized the overall coverage while minimizing the number of fire stations. Therefore, we proposed an optimal layout planning with the conditions of making the most out of the firefighting resources. The optimized location set cover model greatly improved the covered range (Table 4). The coverage rates of the POIs and regional areas reached 97.66% and 84.8%. Additionally, the response time for firefighting can be generally improved in the whole study area (Table 7). Furthermore, the method is very practical. For example, the method proposed by Han [28] uses historical fire data to make decisions on the location of fire stations. This approach may be better suited for cities with longer development histories and more advanced construction, as local departments in such cities have accumulated a large amount of relevant data and can more easily obtain accurate and comprehensive information on the various aspects involved. However, for cities in the initial stages of construction, this method may not be entirely applicable. In such cases, the combined model method can be used to predict the city's fire risk by introducing fire risk points and then determining the location of fire stations to be built accordingly. This approach is reasonable and can help reduce the difficulties caused by insufficient data.

Nevertheless, there are limitations to our model that need to be addressed. The site selection is still influenced by the future development direction of the city, land resources, traffic impedance, population distribution, etc. In other words, an adopted effective approach must adjust the simulation constraints, such as traffic, time, distance, and buildable land, according to the above factors. Significantly, various levels of fire stations are in fact subjected to differentiation in scales and service requirements. However, the assumption was that all facilities were class I fire stations in this study. Therefore, the problems existing in the simulated results should be fully considered in actual applications.

Regarding future research, the current work can be extended in several aspects. With the development of the digitization of cities, there is an opportunity to obtain more accurate and real-time data, which should be integrated into the ArcGIS database to optimize fire station planning. Meanwhile, as mentioned before, the method did not take into account the properties of the fire station, once various simulated construction targets can be employed in the model, the reliability of fire station planning can be increased. Furthermore, by means of gradually screening and integrating the critical influencing factors, combined with data fusion technology, the follow-up research on fire station layouts will be improved gradually.

## 5. Conclusions

In this study, a systematic optimization approach for urban fire station planning based on the combined model method was proposed and applied in Fengdong New Town, Shaanxi province, China, which is currently undergoing rapid development but lacks adequate fire service. The analysis of the variations in POI and area coverage rates showed that the combined model method provided a solution for spatial decision-making by determining the optimal site locations for fire stations and the necessary number of fire stations to meet the requirements. The results revealed that 11 ordinary class I fire stations should be added to Fengdong New Town in the future. The application of the combined model method resulted in an increase of 10.20% and 12.43% in fire station proximity point coverage for POIs and areas, respectively. The optimized fire station layout achieved POI coverage of 98.4% and regional area coverage of 96.8% within a 4 min of fire response time, considering only medium-high-risk sites in the study area. Despite some limitations, the results of this model can provide significant and calculated information for the more effective allocation of future fire stations, which is a crucial issue for the local government and fire department.

**Author Contributions:** Zhijin Yu: Conceptualization, methodology, validation, formal analysis, investigation, resources, writing—review and editing, funding acquisition. Lan Xu: Investigation, validation, formal analysis, writing—original draft, data curation. Shuangshuang Chen: Investigation, methodology. Ce Jin: Investigation, methodology. All authors have read and agreed to the published version of the manuscript.

**Funding:** This research was funded by the Natural Science Basic Research Program of Shaanxi (Program No. 2022JQ-408).

**Data Availability Statement:** The original data were supported by network data.

**Conflicts of Interest:** The authors declare no conflict of interest.

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
