# Peer review of "Research on Urban Fire Station Layout Planning Based on a Combined Model Method"

_ijgi, doi:10.3390/ijgi12030135_

Round 1
Reviewer 1 Report (Previous Reviewer 3)
The rewriting improved the readability of the methodology and it's a little bit easier to follow and reproduce now. If on the one hand I still think the article needs a major overhaul to become truly readable, on the other hand I recognize that the methodology is in itself lengthy and complex, so there is probably no simple way to describe it.
Nonetheless, the methodology is imaginative and has merit, so it will probably catch some attention within academia. Outside it, authors would need to wrap it in a "blackbox" software package to be used by municipal planners, as I don't think practitioners will be willing to learn and implement the methodology from scratch.
Author Response
We appreciate your recognition of the merit of our methodology. We have made efforts to improve its clarity and conciseness, and will continue to work on enhancing the article's readability. We will take your comments into consideration to explore the possibility of developing a software package that practitioners could utilize in the future.
Reviewer 2 Report (Previous Reviewer 2)
Before laying out the comments for the peer review of the manuscript entitled “Research on urban fire station layout planning based on the 2 intersection model method”, I would like to make known that this is the 3rd time when I review this paper. This means that I am familiar with both its improvements, and its shortcomings.
I recommend (like I did previously), major revision based on the following reasons:
- The Introduction lacks a clear statement of the aim of the research, as well as its implications at national and international level. This comes at the end of Introduction and should be followed by a paragraph with the contribution of the paper to the research field. Also, the Introduction should make clear if the applied model was developed by the authors, or by someone else, or if it was adapted to the setting specific to China.
- The caption of Figure 1 needs to be improved. “Data” is vague. The map shows the location of fire stations, the road network configuration and the urban boundary. Also, the left map (with China) lacks a graphic scale.
- Subsection 2.2., the source of the data should be clearly stated, not provided like “Internet”.
- I appreciated that the authors shortened the Methodology, but they have done so by augmenting the Results with subsections that are related to the methodological framework, and not the findings (Subsection 3.1). The Methodology section would rather benefit from a workflow diagram, and the subsections move to Results should be trimmed and included in Methodology.
- Results: The findings of the paper start to be actually presented at line 311.
- My main concern is with the Discussion section. I repeatedly provided guidance on how to construct this very important part of a scientific paper. Right now, the section only critically go over the findings. Discussion should go further, and compare the findings with previous ones form the literature – if not from the same place (obtained through other models), for other places in China or in the world (obtained through the same model or from similar/different ones). This will place the findings in a broader context, and lead to the next paragraph about the implications of the results. What are they useful for, what does this model contribute to the research field? Finally, a paragraph with the limitations of the paper should be included in the Discussion – there are some mentioned about this, but the phrase should clearly start with The limitations are…. (or something similar, of course).
- I also recommend the authors to have their manuscript checked for English language grammar and style mistakes.
Author Response
Reviewer |
|
Comments to the Author |
Response |
1. The Introduction lacks a clear statement of the aim of the research, as well as its implications at national and international level. This comes at the end of Introduction and should be followed by a paragraph with the contribution of the paper to the research field. Also, the Introduction should make clear if the applied model was developed by the authors, or by someone else, or if it was adapted to the setting specific to China.
2. The caption of Figure 1 needs to be improved. “Data” is vague. The map shows the location of fire stations, the road network configuration and the urban boundary. Also, the left map (with China) lacks a graphic scale.
3. Subsection 2.2., the source of the data should be clearly stated, not provided like “Internet”.
4. I appreciated that the authors shortened the Methodology, but they have done so by augmenting the Results with subsections that are related to the methodological framework, and not the findings (Subsection 3.1). The Methodology section would rather benefit from a workflow diagram, and the subsections move to Results should be trimmed and included in Methodology.
5. My main concern is with the Discussion section. I repeatedly provided guidance on how to construct this very important part of a scientific paper. Right now, the section only critically go over the findings. Discussion should go further, and compare the findings with previous ones form the literature – if not from the same place (obtained through other models), for other places in China or in the world (obtained through the same model or from similar/different ones). This will place the findings in a broader context, and lead to the next paragraph about the implications of the results. What are they useful for, what does this model contribute to the research field? Finally, a paragraph with the limitations of the paper should be included in the Discussion – there are some mentioned about this, but the phrase should clearly start with The limitations are…. (or something similar, of course).
6. I also recommend the authors to have their manuscript. |
1. We thank the reviewer’s comments, We have rewritten the closing paragraph based on your suggestion. The changes are highlighted in red in the text.(Line 109-124)
2. We thank the reviewer’s careful check in this paper, about this question. We changed the title to "The geographical location of Fengdong New Town and its location and spatial distribution in the region" and added the missing part of the map on the left.
3. We thank the reviewer’s comments. it is explained in more detail below how the data were obtained. However, the passage is indeed worded incorrectly. We have rewritten the passage.(Line 146-148)
4. Thank you for your feedback on our article. We appreciate your suggestion regarding the Methodology section and will take it into consideration as we continue to work on improving the readability of the article. A workflow diagram is a great idea to better visualize the methodology, and we will work on incorporating that into the section. We will also review the Results section and make sure that any subsections related to the methodology are included in the Methodology section and not the Results section. Thanks again for your helpful comments.
5. Thank you for your expert opinion. Your main concern is with the Discussion section of the scientific paper, and you have provided guidance on how to construct this important part. We have added some new elements to the discussion based on your suggestions and have marked them in red in the text. (Line 412-422)
6. We thank the reviewer’s comments. Thank you very much for your kind suggestions. We have carefully checked the full text grammar and made careful corrections. |

Reviewer 3 Report (New Reviewer)
This paper presents a new method for locating fire stations, which combines existing optimization methods for fire station layout. The authors propose a fire station location model based on ArcGIS and multi-objective decision making, and demonstrate its effectiveness through a case study. However, the research presented in this paper is innovative and applicable, there remain some aspects that could benefit from further refinement.
1. The location-allocation model utilized in this paper is not limited to the maximum coverage model, and therefore it is recommended to include a keyword such as "location model" to better reflect its scope.
2. What is the relevance of the Layout principle to the article? The paper should be concise and it is recommended to simplify the language and reduce useless exposition and do not simply list it.
3. Where does the formula A=2P2 in Section 3.2 come from? Please explain it clearly.
4. Please correct the units in the“Coverage amount”column in Form 4, as some units were written and some were not.
5. In Section 2.1, you wrote that" while the number of existing fire stations is far below the real demand.". What is the basis for that judgment?
Author Response
Reviewer |
|
Comments to the Author |
Response |
1. The location-allocation model utilized in this paper is not limited to the maximum coverage model, and therefore it is recommended to include a keyword such as "location model" to better reflect its scope.
2. What is the relevance of the Layout principle to the article? The paper should be concise and it is recommended to simplify the language and reduce useless exposition and do not simply list it.
3. Where does the formula A=2P2 in Section 3.2 come from? Please explain it clearly.
4. Please correct the units in the “Coverage amount” column in Form 4, as some units were written and some were not.
5. In Section 2.1, you wrote that" while the number of existing fire stations is far below the real demand.". What is the basis for that judgment? |
1. Thank you for your suggestion. We agree that including a keyword such as "location model" would better reflect the scope of our methodology. We will take this into consideration and make the necessary changes in the paper.
2. Thank you for your feedback. The Layout principle is indeed an important component of the proposed methodology, as it aims to optimize the location of fire stations to achieve the best possible coverage. In our revised version of the article, we will work on improving the language and structure to make the explanation of this principle more accessible to readers. Thank you for your suggestion.
3. We thank the reviewer careful checked on this paper. This formula is an empirical equation frequently utilized to estimate the protective radius of a fire station, based on the station's assigned area of responsibility.
4. We thank the reviewer careful checked on these figures. We have corrected the units of the table data.
5. We thank the reviewer careful checked on this paper. Thank you for your inquiry. Our assessment is based on the results of our on-site visits and research. We found that the current number of fire stations falls short of meeting the actual demand, which is further supported by the "Urban Fire Station Construction Standards". Thus, we concluded that there is a need to increase the number of fire stations. We will make sure to clearly express this point in our paper. Thank you again for your guidance and suggestions. |

This manuscript is a resubmission of an earlier submission. The following is a list of the peer review reports and author responses from that submission.
Round 1
Reviewer 1 Report
The article presents a scientific proposal that can make a real contribution to improving community safety by optimising the location of fire stations. The research is scientifically sound and complementary to previous research in this field. The research design is appropriate. Overall, the clarity of the presentation is good, but a few corrections and additions would be helpful. These relate to the following issues:
- lines 25-32 - provided data should be referred to,
- lines 47-48 - the names should be noted in the references,
- line 256 - why are such parameters given? Are they embedded in specific regulations, or is this a selection proposed by the authors?
- Figures 8, 9, 14, 15 and 16 - the legend and (in some cases) the scale scales overlap the map.
In addition, minor composition errors can be found in the text - e.g. line 405.
Author Response
Comments to the Author |
Response |
1. Lines 25-32 - provided data should be referred to.
2. Lines 47-48 - the names should be noted in the references.
3. Line 256 - why are such parameters given? Are they embedded in specific regulations, or is this a selection proposed by the authors?
4. Figures 8, 9, 14, 15 and 16 - the legend and (in some cases) the scale scales overlap the map. |
1. We thank the reviewer’s comments, we add references to the data. (Ln25-32)
2. We thank the reviewer’s careful check in this paper, about this question. The names are noted in the references. (Ln52)
3. We thank the reviewer’s comments. This data is set by the author with reference to the appropriate slope of building construction in other cities, and it is more convenient for fire vehicles to enter and exit under this terrain slope. 4. We thank the reviewer’s careful check in this paper. We have adjusted the size scale of the legend appropriately in Figures 8, 9, 14, 15 and 16. |
Reviewer 2 Report
The paper called “Research on urban fire station layout planning based on intersection model method” provides an engaging reading, although there are some writing style mistakes that need to be corrected. I advise you ask a native speaker to go through the manuscript and improve it.
The manuscript needs polishing, and most of all restructuring, as right now it appears that it was written by inexperienced scientists. This is not a reason to worry, but a starting point for scientific writing improvement. Also, the paper lacks a Discussion section, the aim statement is confusing and incomplete. These are all reasons that support the major review decision.
Please see the attached pdf for on-point comments with review suggestions. The red boxes are the comments that should have the biggest impact on the improvement of this manuscript, but the green ones also have to be addressed. I hope you will find the review useful and that it will help your scientific progress.

Author Response
Comments to the Author |
Response |
1. Overall, the Abstract is well-written. However, the background info in the beginning is too extended, while the implications of the Results (which should be placed at the end) do not exist. Please amend this issue.
2. These keywords are too general. Try to find more on point keywords, in order to make your research easily discoverable and more impactful.
3. This presentation of the existing literature is appropriate, but it is not followed by statements that relate to the research gap. This research gap should be defined before the aim statement, and the aim should be presented as a solution for the research gap.
4. The Introduction should end with the practical implications of the study. After stating the aim of the study, the authors should explain what is its contribution to the scientific literature and if the results may be used in practice to improve certain situations.
5. The Methodology section (which should be named like it) should start with the Study area, Data and then with the Intersection Model. This section needs massive restructuring.
6. There is no need to present these 4 models which are used in the literature. Only the selected model to be applied needs to be presented in an accessible language. Try to keep it neat and simple. Therefore, this model support section needs to be shortened.
7. This needs to be detailed. As this model is the only one that needs to be presented, a thorough presentation should be provided. Also, the highlighted part needs to be made more clear.
8. (1)The first statement should include the geographic position of the study area in the region and country.(2)This section should include a map with the street network of the study area, the location of the fire stations, and also a location map within the country and region.(3)This section should include a map with the street network of the study area, the location of the fire stations, and also a location map within the country and region.
9. Study area and Data Source should be part of the Methodology section. It does not belong in the Results.
10. The exact source should be provided.
11. What modifications were performed?
12. These layout principles should be part of the Methodology section. As I can see, this is a very particular methodological framework, and the authors should specify in the Discussion section (which is yet to be written) how the framework can be modified to be applied in other cities. Otherwise, this paper is of interest only for regional/national audiances at most.
13. The anaysis of the accessibility to high-risk places is a very strong point of this paper. Unfortunately, the Methodology does not mention this, nor does the aim in the Introduction. Please correct this.
14. This manuscript lacks a Discussion section. This should be placed after Results and before Conclusions. The Discussion section should provide a critical view of the results, a context-based interpretation of the results, details on how these findings fit into the existing literature body, what does this study bring new and useful, what are its practical implications and also its limitations. In fact, this is one of the most important sections of a paper. |
1. We thank the reviewer’s positive comments. In the revised version, we revise the background and modify summary to indicate the objectives of the study and the main contribution.
2. We thank the reviewer’s comments. Based on the comments, we replaced the keywords with “intersection model method; ArcGIS; the nearest facility model; the maximal covering location model; model optimization; crosspoint; coverage rate”.
3. We thank the reviewer’s comments, We added a paragraph about the shortcomings of the current fire station siting model.(Ln95-107)
4. We thank the reviewer’s comments, about this question, We have reworked the end of the introduction.(Ln108-116)
5. We thank the reviewer careful checked on this paper. We have reorganized the structure of the methodological part of the content.
6. We thank the reviewer careful checked on these figures. On this issue, We simplified the model support section.
7. We thank the reviewer’s comments. We have enriched the subsection with a more detailed description of the steps of the intersection model method approach.
8. We thank the reviewer’s careful check in this paper. To address the above issues, we added the geographic location of the study area, fire history data, etc., and added Figure 1 as a graphical representation of the city location and fire station location.
9. We thank the reviewer careful checked on this paper. We have added this part to section 2
10. We thank the reviewer careful checked on this part. Our main data sources are online public information, online map Baidu Open Map, field research, etc. We made changes in the following articles
11. We thank the reviewer careful checked on this paper. There will be more duplicate data in POI data, such as roads, buildings, etc., so this kind of duplicate POI data will be deleted and processed. And there will also be missing coordinate information or wrong functional type classification, correct and complete the POI against the data source, so as to reduce the subsequent urban renewal project decision analysis errors.
12. We thank the reviewer careful checked on this paper. We have included a reflection on this issue in the conclusion and discussion section.
13. We thank the reviewer careful checked on this paper. First, we add a discussion of this content in the introduction. The accessibility analysis of high-risk sites is also mentioned in the methodology.
14. We thank the reviewer’s comments. Based on your suggestion, we have changed the conclusion section to a conclusion and discussion section. The shortcomings of the method and possible improvements have been added in this section. |
Reviewer 3 Report
This article presents a decision support methodology to optimize fire station locations based on a combination of existing models. I have some major issues with this paper:
1. The problem is more relevant to an operations research or disaster relief journal than to IJGI because the geo-information involved is mostly restricted to the dataset and case study implementation.
2. Authors speak broadly of "the intersection model" but do not clarify what its novelty is (compared to the literature) nor what research gap it fills. The point is that many facility location models exist in the literature, as authors themselves recognize in their methods section, so motivation for introducing another one becomes key. Saying that case study results improve the actual situation on the field is not enough, as most alternative methods will also provide some kind of improvement. Authors must show what are the plus-values of their new methodology, as compared to other methodologies.
3. The presentation is insufficient to understand how the methodology works and is implemented. Because the whole procedure from A to Z is lengthy and complex, authors try in section 2.2 to draw a big picture on how it operates. However, they fail miserably because rather than clarifying the idea, that section only succeeds in filling the reader with doubts and questions. In fact, the whole paper is like that: it seems to be written from the authors to themselves, as many crucial implementation details are missing, leaving the reader guessing what authors are actually doing or where they want to get at. Because the methodology and subsequent sections are insufficient to understand how the methodology works, the whole article is rendered pointless because results cannot be reproduced.
I agree some implementation details are best shown in practice, i.e., within case study sections 3 and 4, and from there I can dig out part of the whole idea. However, section 2.2 must absolutely be self-standing: readers should be able to understand the full methodology just by reading this section. Having to go through the whole article a couple of times to try and guess what's going on is not acceptable and greatly diminuishes the article's chance to be cited.
Some points to be addressed are:
Line 152 and 162. Clarify the nature of Aij and its value range.
Line 159. What is the "specified impedance" and what is its effect on the models? Doesn't it need to be added to the linear programming models (nearest facility, max covering, min facility) somehow?
Line 187. Clarify what "intersection point of time and distance" means.
Line 187. The sentence "intersection point of time and distance is obtained using the intersection model method" cannot be used because the intersection model method has not yet been defined!
Line 188. I believe buffer area is the same as service area. If so, make this explicit.
Line 188. Why is buffer area size >= study area condition needed? To make sure non-POI locations are covered by the fire stations?
Line 189. "combining the existing fire stations" What is this and how is it done?
Line 190. "the intersection points are repeatedly created, and so on" Again, what does this mean and by what process do they get created?
And more:
Why not give a map of candidate locations? How are candidate locations eliminated? How are the p-median models (nearest facility, maximal covering), which have a fixed number of candidate locations, used to navigate in the sea of combinations of candidate locations?
I'm pretty sure all these questions have simple answers; yet authors provide little clues to how it's done. I tried to read the thing over and over again and found myself exhausted of trying to guess the author's ideas and methods. I certainly don't want the journal readers to through the same (painful) experience.
Author Response
Comments to the Author |
Response |
1. The problem is more relevant to an operations research or disaster relief journal than to IJGI because the geo-information involved is mostly restricted to the dataset and case study implementation.
2. Authors speak broadly of "the intersection model" but do not clarify what its novelty is (compared to the literature) nor what research gap it fills. The point is that many facility location models exist in the literature, as authors themselves recognize in their methods section, so motivation for introducing another one becomes key. Saying that case study results improve the actual situation on the field is not enough, as most alternative methods will also provide some kind of improvement. Authors must show what are the plus-values of their new methodology, as compared to other methodologies.
3. The presentation is insufficient to understand how the methodology works and is implemented. Because the whole procedure from A to Z is lengthy and complex, authors try in section 2.2 to draw a big picture on how it operates. However, they fail miserably because rather than clarifying the idea, that section only succeeds in filling the reader with doubts and questions. In fact, the whole paper is like that: it seems to be written from the authors to themselves, as many crucial implementation details are missing, leaving the reader guessing what authors are actually doing or where they want to get at. Because the methodology and subsequent sections are insufficient to understand how the methodology works, the whole article is rendered pointless because results cannot be reproduced.
4. Line 152 and 162. Clarify the nature of Aij and its value range.
5. Line 159. What is the "specified impedance" and what is its effect on the models? Doesn't it need to be added to the linear programming models (nearest facility, max covering, min facility) somehow?
6. Line 187. Clarify what "intersection point of time and distance" means.
7. Line 187. The sentence "intersection point of time and distance is obtained using the intersection model method" cannot be used because the intersection model method has not yet been defined!
8. Line 188. I believe buffer area is the same as service area. If so, make this explicit.
9. Line 188. Why is buffer area size >= study area condition needed? To make sure non-POI locations are covered by the fire stations?
10. Line 189. "combining the existing fire stations" What is this and how is it done?
11. Line 190. "the intersection points are repeatedly created, and so on" Again, what does this mean and by what process do they get created?
12. Why not give a map of candidate locations? How are candidate locations eliminated? How are the p-median models (nearest facility, maximal covering), which have a fixed number of candidate locations, used to navigate in the sea of combinations of candidate locations?
|
1. We thank the reviewer’s suggestion. In this paper, we propose a location decision model based on existing research, and the paper has some relevance to spatial anlysis and decision making, so we chose this journal.
2. We thanks the reviewer’s comments, about this question, We enrich our view of existing fire station siting models and refine the innovative nature of the model intersection method in the introduction section.
3. We thank the reviewer’s comments. We supplemented the implementation process of the model intersection method.
4. We thanks the reviewer’s comments. We changed the description of the formula.
5. We thank the reviewer’s suggestion;Impedance refers to time impedance and distance impedance. According to the requirements of urban fire station construction, time and distance impedance are set in ArcGIS. We described this in Section 2.4.
6. We thank the reviewer’s comments.We made changes in the article.
7. We thanks the reviewer’s comments. We have revised the sentence.
8. We thanks the reviewer’s comments. Buffer area is the same as service area, we explain this in our article.
9. We thanks the reviewer’s comments. About this question,The response range of the fire station can be determined through buffer analysis. Therefore, we simulate the ideal state that the buffer area is larger than the study area, that is, the response range of the fire station can cover the whole study area.
10. We thanks the reviewer’s comments."combining the existing fire stations" means All the positions of fire stations obtained by several simulations are taken as the required points of fire stations.
11. We thanks the reviewer’s comments. First, a buffer zone analysis of the city's existing fire stations was conducted to identify candidate fire station sites. After the first use of the model intersection method, the first group of fire station required points can be determined. Then, the buffer zone analysis of the city's built fire stations and the required points of fire stations is carried out again, and the model intersection method is used again to determine the second group of required points from the candidate points of fire stations and so on.
12. We thanks the reviewer’s comments.(1) We added Figure 3 to show the candidate area and candidate point of the fire station. (2) First, points beyond the intersection of the maximum coverage model and the minimum facility point model will be eliminated, and second, when adding more fire station points basically fails to improve POI coverage and area coverage, the remaining fire station candidate points will be eliminated (3) The answer to the third question is the same as the reply to the Article 12 opinion. |
Round 2
Reviewer 2 Report
This is the second time I review the paper intitled “Research on urban fire station layout planning based on the intersection model method“.
The authors have improved the manuscript, but certain comments from the previous were not properly addressed:
- The implications of the Results (which should be placed at the end of the Abstract) do not exist. Please amend this issue.
This comment was not properly addressed, as the end of the Abstract still does not include the implications of the study. Also, in the Abstract the formulation “Taking the Fengdong New Town of Xi’an, China as the background” should be avoided, and replace with one referring to a study area/case study.
- The authors augmented the keyword list, but not to the benefit of the paper. For instance, the method should not be the first keyword, ArcGIS software is not a proper keyword. In order to help, I propose this list and leave it at the choice of the authors: urban fire hazard, urban fire station layout planning, intersection model method, maximal covering location model
- The authors properly identified the research gap, this is ok.
- The Introduction should end with the practical implications of the study. After stating the aim of the study, the authors should explain what is its contribution to the scientific literature and if the results may be used in practice to improve certain situations.
The authors added 1-2 paragraphs at the end of the Introduction, but they barely discuss the novelty of the paper and what its real contribution to the field. These should be thoroughly argued, in order convince the audience (and the reviewers) that the manuscript adds value. Please state the national and international implications of this study.
- I appreciate the changes in the Methodology section. Indeed, it is better now.
- Right now, the paper does not have a Results section. Please combine 3 and 4 (it is my understanding that these constitute the Results) under this title. The names of 3 and 4 can be subsections of the Result.
- The paper must have a separate Discussion section. In my last review, I offered guidance on what should be included in such a section. It is very important to provide interpretations for the obtained results, not just to analyse them. Also, it is important to compare your findings with previous ones in China or other parts of the world. Right now, the manuscript is not anchored in the scientific literature, but drifts aimlessly, although it has value. Please address this issue properly.
This manuscript lacks a Discussion section. This should be placed after Results and before Conclusions. The Discussion section should provide a critical view of the results, a context-based interpretation of the results, details on how these findings fit into the existing literature body, what does this study bring new and useful, what are its practical implications and also its limitations. In fact, this is one of the most important sections of a paper.
The Conclusion section should provide a critical view of the research field, the place of your results in this field, future research perspectives etc.
I hope the authors will find these comments useful and that they will improve the paper.
Reviewer 3 Report
While I appreciate the author's efforts in clarifying the methodology and its implementation details, the article remains too difficult to both follow. New lines 247-263 help understanding the idea, but there is simply too much going on and authors fail to present all the steps in an intelligible and consistent way. The text is a nightmare to read, with new things constantly being thrown into the mix, demaning unrelenting attention for hours. Quite a bit of going back and forth is required in the reading, and partial conclusions are derived that are difficult to fit in the grand picture. In short, and as I have already mentioned in my first revision, the article was written by the authors for the authors, not for the audience of IJGI. Sadly, it remains so.
I am sorry if this sounds a bit harsh, but truth is every time I try to read the article I have a really hard time going through it, let alone reproduce the methodology. Perhaps I'm being too demaning, but I firmly believe the merit of an article goes beyond its technical prowess: to be truly impactful, an article must be readable. Otherwise it no one will care.
Anyway, there is nothing more specific that I can add. Should the article receive the green light for publication in its present state, I only recommend revising the English language, as many sentences are incorrectly formulated and grammar issues are plentiful.